# Evaluation of the Effect of SPIDER System Therapy on Weight Shifting Symmetry in Chronic Stroke Patients—A Randomized Controlled Trial

**DOI:** 10.3390/ijerph192316214

**Published:** 2022-12-04

**Authors:** Paulina Magdalena Ostrowska, Rafał Studnicki, Marcin Rykaczewski, Dawid Spychała, Rita Hansdorfer-Korzon

**Affiliations:** 1Department of Physiotherapy, Medical University of Gdańsk, 7 Dębinki Street, 80-211 Gdańsk, Poland; 2Second Department of Radiology, Medical University of Gdańsk, 17 M. Smoluchowskiego St., 80-214 Gdańsk, Poland

**Keywords:** stroke, gait disorder, weight shift, balance, SPIDER system

## Abstract

(1) Background: The Strengthening Program for Intensive Developmental Exercises and Activities for Reaching Health Capability (SPIDER) system is dedicated to patients with motor deficits resulting from damage to the peripheral or central nervous system (including post-stroke patients). It enables the conduct of forced-weight-bearing therapy to the lower limb affected by the paresis. In this study, the TYMO^®^ measuring platform was used to quantify the impact of therapy using the SPIDER system and therapy that did not use this system. The TYMO^®^ device is a portable posturography platform that monitors the tilting of the body’s center of mass and reports the results of the rehabilitation process. (2) Objective: To evaluate the effect of therapy based on neurophysiological methods (proprioceptive neuromuscular facilitation (PNF), neurodevelopmental treatment according to the Bobath concept (NDT-Bobath)) and the SPIDER system on body weight transfer shifting, in post-stroke patients in the chronic phase, compared to therapy based on neurophysiological methods (PNF, NDT-Bobath), without the use of the SPIDER system. (3) Methods: This is a randomized controlled trial in which patients (*n* = 120; adults, post-ischemic stroke—first stroke episode, in chronic phase—up to 5 years after the stroke incident) were assigned to one of two groups: study, *n* = 60 (with therapy using PNF, NDT-Bobath methods and the SPIDER system); and control, *n* = 60 (with therapy using PNF and NDT-Bobath methods, without the SPIDER system). In patients in both groups, before and after the training (2 weeks of therapy), body weight distribution was measured on the TYMO^®^ platform. (4) Results and Conclusions: The results of the statistical analysis demonstrated a greater reduction in the tilt of the body’s center of mass in therapy using the SPIDER system, compared to therapy in which the system was not used.

## 1. Introduction

Of the 15 million people worldwide who suffer a stroke each year, 5 million are permanently disabled and need ongoing care [1]. Improving post-stroke patients is a very complex and difficult process. Due to a combination of motor, sensory and cognitive disorders, a large percentage of post-stroke patients experience imbalance and limitations in activities of daily living [2,3,4]. Tyson et al. found that 83% of patients report balance problems, which are usually observed as asymmetry in weight shifting towards the unaffected side of the body [2,5]. Other studies have also confirmed the concept of shifting the center of mass toward the unaffected side [6,7,8]. In addition, imbalance and reduced ability to carry weight symmetrically are assumed to be contributing factors to falls in post-stroke patients [2,9]. A study by Patterson et al. found that 55.5% of chronic stroke survivors demonstrate gait asymmetry [10]. An asymmetrical gait pattern not only increases one’s energy expenditure or risk of falls, but can also cause bone density loss in the affected lower limb, increasing the risk of musculoskeletal degenerative changes and the onset of pain in the unaffected limb (due to prolonged weight bearing) [11,12]. As early as 4 to 30 h after a stroke incident, a decline in the number of motor units in skeletal muscle tissue begins. Loss of muscle mass is associated with impaired neurovegetative control, a decrease in the number of motor neurons, degeneration of neuromuscular connections, systemic catabolic–anabolic imbalance and local metabolic changes in muscle. Muscle weakness (a reduction in maximal muscle contraction) is observed in both the limbs of the affected and unaffected side. In the affected limb, the composition of the muscle tissue is changed; there is an increase in intramuscular fat relative to muscle tissue, which is inversely related to the level of oxygen consumption during peak effort. In post-stroke patients, there is also a shift from slow-twitch to fast-twitch muscle fibers, and this physiological compensatory mechanism is seen in conjunction with a greater severity of gait deficits [13]. The aforementioned asymmetry is a result of muscular imbalance; the muscle tone of the affected side decreases, causing an increase in the muscle tone of the unaffected side. Three months after the stroke incident appears the increased pathological muscle tone of the affected side (spasticity) [14]. A weakening of the muscles of the affected side and the appearing spasticity of these muscles is the main reason for the increase in the frequency of weight shifting to the unaffected side. The lack of weight bearing on the affected side, in turn, generates an additional increase in the spasticity of the hemiparesis side and increases the weakness of the muscles of this side [6,15,16]. Therefore, various forms of improvement are being searched for post-stroke patients, emphasizing the return of the hemiparesis side’s motor abilities identical or as similar as possible to physiological movement patterns. The forced use of the lower limb of the hemiparesis, as a therapeutic tool during physiotherapeutic treatment, can influence higher tolerance of the patient’s postural correction and lead to an improvement in balance and gait symmetry, and therefore its economy and ergonomics [16,17]. For this reason, it is highly important in the process of improving the post-stroke patient to facilitate the transfer of body weight to the directly affected side. The present study attempts to use the SPIDER system to improve the symmetry of body weight shifting in post-stroke patients in the chronic phase, which is the main component of therapy to help return to the physiological gait stereotype.

The above-mentioned topic of imbalance and asymmetric gait is related to the investigation of the pattern of walking (gait analysis). Clinicians use qualitative analysis and observation-based balance scales (e.g., Berg Balance Scale, Trunk control test, TUG test) because these tests do not require equipment input and are easy to perform. However, quantitative analysis is more accurate and provides more information about the patient’s functional status. Posturographic platforms are used to quantitatively measure the center of the body’s mass tilts in standing, which interpret standing stability and postural control strategies [18,19]. Common methods of gait analysis include using cameras to track the position of the body by using reflective markers. An alternative is to use acceleration and angular velocity data measured from inertial sensors attached to the body. Currently, several applications in human motion analysis may benefit from miniature inertial sensors [20]. These methods are expensive, require the maintenance of a dedicated motion laboratory and use cumbersome equipment attached to the patient. Another option is a wireless system that allows data collection throughout the day in a variety of environments (not just laboratory conditions): the ‘gait shoe’. This system has been designed with components configured to minimally affect gait, and is readily fixed on typical athletic shoes. It is capable of sensing many parameters that characterize gait: length and velocity of stride, orientation, force distribution under foot, heel-strike timing and toe-off timing, plantar-flexion and dorsiflexion, height of foot above the floor [21].

## 2. Materials and Methods

### 2.1. Subject of the Study

This is a randomized controlled study approved by the Independent Bioethics Committee for Scientific Research at the Medical University of Gdańsk (consent of 7 July 2021, Resolution No. NKBBN/507/2021), in cooperation with the NORMAN Neurological Rehabilitation Center in Koszalin. The study was conducted from August 2021 to August 2022. From a group of all post-stroke patients (*n* = 300) residing at the center at that time, patients meeting the inclusion criteria were selected (*n* = 131:*n* = 120—patients were randomly allocated to one of two groups: study and control; *n* = 11—patients did not receive the allocated intervention) (Figure 1). All patients participating in the study were informed in detail about the nature and procedure of the study and gave written voluntary agreement to participate. The study group (SG) includes 60 patients treated with neurophysiological special methods (PNF, NDT-Bobath; 2 weeks of therapy, 4 × 45 min daily) and the SPIDER system (2 weeks of therapy, 1 × 45 min daily). The control group (CG) includes 60 patients treated with neurophysiological special methods (PNF, NDT-Bobath; 2 weeks of therapy, 5 × 45 min per day), without using the SPIDER system.

### 2.2. Assessment Protocol

Inclusion criteria (Table 1) were determined on the basis of a questionnaire form using the following scales and tests: a questionnaire with functional tests based on the ICF, the Barthel Scale, the Rankin Scale, the NIHSS Scale, the Modified Ashworth Scale, the FMA-LE test, the mirror test and the Berg Balance Scale, to guarantee that the groups were as homogeneous as possible. The patient’s diagnosis was completed by a doctor; tests evaluating the patient’s functional status were assessed by a physiotherapist.

ICF-based functional testing:
-Change from sitting to standing (d4103)—the patient performs a transition from sitting with feet placed on the floor to standing (without any assistance);-Holding the standing position (d4154)—the patient stays in the required standing position for a specified period of time (without any assistance);-Test to assess the risk of falls in walking (TUG—d4500)—the patient stands up from a sitting position (e.g., on a chair) and walks a distance of 3 m, then turns around and returns to the starting point by sitting down again. The time is measured from standing up to taking up a sitting position again. Norms: healthy people—less than 10 s, elderly people—14 s, patients at risk of falling—more than 30 s [22,23].Modified Ashworth Scale: a six-point numerical scale used to assess increased muscle tone (spasticity). Muscle tension in the lower limbs is tested in the supine position, symmetrically on both sides. The speed of passive movement, which should last approximately 1 s in full flexion or extension, proves to be important [14,24].Barthel scale: an international scale for assessing a patient’s functional abilities and care needs. The index assesses the patient’s level of dexterity and independence in ten basic areas: eating, moving and sitting, maintaining personal hygiene, using the toilet, washing and bathing the entire body, moving on flat surfaces, going up and down stairs, dressing and undressing, controlling feces and anal sphincter, controlling urine and bladder sphincter. Interpretation of scores: 0–20 pts.—complete dependence, 21–80 pts.—help from others needed, 81–100 pts.—patient can function independently with support from others [25].Rankin scale: a six-point scale assessing the level of disability of a neurological patient (where “0”—patient has no complaints, “5”—very severe level of disability, the patient is completely dependent on the environment, constant assistance from another person is necessary) [25,26].NIHSS scale: an international scale to assess the state of consciousness, orientation, response to commands, associated gaze, visual field range, facial and limb muscle paresis, sensation and language function of the stroke patient. Scores range from 0 (normal) to 42 points (where 1–4—minor stroke, 5–15—moderate stroke, 16–20—moderate to severe stroke, 21–42—severe stroke) [27].MMSE scale: a mental status evaluation scale that allows quantitative assessment of many aspects of cognitive function: orientation in time and place, memory, attention and counting, remembering, language function, repetition, following complex oral or written instructions and visuospatial ability. The maximum score that can be obtained in the test is 30 points (a score below 24 points is indicative of varying degrees of Dementia) [28].FMA-LE test: an index to assess lower limb sensorimotor dysfunction in post-stroke patients. The scale includes an assessment of: reflex activity, voluntary movements within and beyond synergies, ability to perform isolated movements, speed and coordination. A total score of 34 points indicates normal lower limb function [29,30].Mirror test: a test used to investigate deep sensation [31].Berg Balance Test: a quantitative test to assess balance and fall risk. It focuses on static and dynamic balance. It includes 14 tasks with a maximum score of 56 points. A score of 0–20 points suggests total wheelchair dependence; 21–40 points are assigned to a patient who moves with assistance, 41–56 points—independent patient [29,32].

### 2.3. Physiotherapy Treatment Program

The physiotherapeutic treatment of both groups was based on manual therapy, balance training and postural control, sensory and functional techniques using neurophysiological methods (PNF, NDT-Bobath), and exercises on equipment (stationary bicycle, treadmill). In addition, the study group was treated using the SPIDER system. Before and after two weeks of the rehabilitation program, active body weight distribution (COFT) analysis was performed on the TYMO^®^ platform. The study group’s training lasted 2 weeks: 4 × 45 min per day of therapy based on neurophysiological methods and 1 × 45 min per day of therapy using the SPIDER system. In contrast, training of the control group lasted 2 weeks: 5 × 45 min per day of therapy based on neurophysiological methods, without the use of the SPIDER system.

#### 2.3.1. TYMO^®^ platform

TYMO^®^ is a device for analyzing the transfer of the body’s center of mass; it compares the symmetry of the active weight distribution and quantitatively assesses balance and postural control in the standing position (Figure 2). The TYMO^®^ analysis includes information on:-The length of time (in seconds) that balance can be maintained;-The range of motion (ROM) exceeded;-The center of force track (COFT), or the total length of the center of force track (COF) during each measurement. Medio-Lateral indicates the total range of lateral sway of the body (right-mid-left), and Anterior-Posterior indicates the total range of forward and backward sway of the body (front-mid-backward);-The COF movement area (the ellipse area, which contains 95% of all COF positions);-The Stability Dynamics Index (STDI), which analyzes the relationship between the distance traveled by the COF and the area of ROM. The higher the value, the more unstable the patient;-An average speed and the ratio of maximum speed to average speed. The speed value describes the number of fast movements that need to be made to maintain balance;-A posture, viz., the distribution of weight and/or force from front to back and left to right [33].

Each patient, before therapy and after two weeks of training, was given the task of maintaining a standing position on the TYMO^®^ platform (standing barefoot, with eyes open). Prior to the test, each patient was instructed to distribute their body weight symmetrically on both lower limbs during the test. The test of maintaining the above-described position lasted 30 s (Figure 3). The device recorded the tilts of the body’s center of mass in a specific direction. On the basis of the results obtained (the difference in the size of tilt of the center of mass, measured in centimeters, before and after the physiotherapy treatment), the effect of therapy with and without the SPIDER system on the symmetry of body weight transfer to the affected and non-affected lower limb was assessed.

#### 2.3.2. SPIDER System

The SPIDER device, as part of the overall system (metal cage), is based on elastic cords, which are the stimulating elements. The cords are attached to a support belt, which in turn is placed around the patient’s waist. The force generated depends on the type of expander (two types of cords are used) and the height of attachment on the cage of the SPIDER device (height of expander attachment—center of gravity angle) (Figure 4) [34,35,36].

The advantage of the SPIDER system is the ability to work with the patient in a standing position. The patient is off-loaded if the point of attachment of the cords to the cage is above the level of the support belt (above the transverse plane and therefore above the center of mass). Conversely, the patient is loaded if the cords are attached below the level of this belt (below the transverse plane). This quick way of changing the load and unloading is a useful possibility in the training [34]. In patients who have a problem with learned disuse of the lower limb of the affected side, transferring body weight to this limb is challenging. Therapy with the SPIDER system in off-loaded conditions allows the patient to gradually shift their body weight to the side directly affected, thus minimizing the generation of associated reactions or compensations. In contrast, patients who do not have the phenomenon of learned disuse are improved under load conditions. In this way, the activated muscle groups are exposed to additional work, while the increased pressure (approximation) of the joint surfaces improves deep sensation and generates an increase in the tension of the weakened muscles [17,34].

In the present study, therapy using the SPIDER system consisted of forcing the body weight onto the affected lower limb to inhibit the mechanism of learned disuse of this limb (Figure 5 and Figure 6). This resulted in muscle activation of the weight-bearing lower limb, an increase in muscle tone of the hemiparesis side, an improvement in sensation (both: superficial and deep) in the foot, and a prolongation of the support phase of the hemiparesis lower limb, all of which were reflected in improved symmetry of weight shift in gait. Photo 5. shows a patient with right-sided paresis: the patient shifted his body weight to the directly affected side. The SPIDER system, via strings attached to the belt, provided him with stabilization in the forced position. The therapist facilitated active positioning of the affected lower limb, viz. slight flexion of the right knee joint to prevent it from locking on passive structures (in a hyperextension), while at the same time the patient, with the unaffected lower limb positioned in abduction at the hip joint, performed flexion and extension (mobility on stability). Photo 6 shows a patient with right hemiparesis, placed in a forced loading position of the affected lower limb. The left lower limb performed flexion, abduction and external rotation movements (PNF pattern) during this time. The therapist applied resistance to the right hip joint to additionally stimulate the patient to shift weight to the right side while making sure that the patient did not position the right knee joint in a hyperextension. By crossing the upper limbs over the chest, the patient provided even more trunk stabilization. In both cases, the patient is off-loaded; the ropes are positioned above the level of the weight-bearing belt.

### 2.4. Statistical Analysis

Tests of normality were performed using the Shapiro–Wilk Test. These tests did not indicate the normality of the distribution. However, since *n* > 30, the *t*-test could be used (robustness of the test for non-compliance with the normality condition). Indeed, the t distribution can be approximated by a normal distribution because the sample size in this case (*n* = 60) is large. Descriptive statistics were performed using the Student’s *t*-test. Data management and analysis were performed in two programs: Microsoft Excel 2007 (Microsoft: Redmond, WA, USA) + add-ons and StatSoft Statistica V12 Advanced Package (StatSoft: Tulsa, OK, USA). The statistical significance level was set at *p* < 0.05 (one-tailed).

## 3. Results

### 3.1. Demographic and Functional Characteristics of Patients

The final study sample included 120 participants, of whom 31.67% were female and 68.33% were male, ranging in age from 24 to 84 years, with a mean of 56.13 years (SD = 14.4) (Table 2). According to random allocation, the SG group comprised 60 patients receiving therapy using the SPIDER system, and the CG group comprised 60 patients receiving therapy without the SPIDER system. There were no significant differences in demographic and functional values between the groups (for all the variables: *p* > 0.05) (Table 3).

### 3.2. Description of Results

Expressed in centimeters, the mean tilt of the body’s center of mass, measured with the TYMO^®^ platform, after rehabilitation with the SPIDER system (2.14 cm) is significantly smaller than the mean tilt before rehabilitation (4.29 cm), measured in the same way (the t Stat parameter is less than 0 (−13.6542), and P(T ≤ t) one-sided (3.22093 × 10^−20^) is less than 0.000001 and is therefore smaller than the assumed α = 0.05) (Table 4).

Figure 7 shows a comparison, in the form of a box plot, of the statistical analysis distribution of the patients’ center of mass (measured in centimeters) before (B) and after therapy (A) using the SPIDER system. This graph illustrates the difference in median, minimum and maximum values. The minimum and maximum values for Boxes A and B are in different ranges; they do not overlap, indicating a large reduction in the tilt of the body’s center of mass after therapy. A difference of 50% (2.15 cm), between the mean value of tilt before rehabilitation (Box B: median = 4.29 cm) and after rehabilitation (Box A: median = 2.14 cm) also indicates a significant reduction in tilt of the body’s center of mass. The range of Box A (0.7 cm) is smaller than the range of Box B (1.2 cm), indicating a reduction in the disparity of the tilt of mean center of mass results for the SG group after rehabilitation with the SPIDER system.

The values of the mean tilts of the body’s center of mass from the measurements on the TYMO^®^ platform before (3.55 cm) and after rehabilitation (3.47 cm) using therapy without the SPIDER system are close to each other; the result is not statistically significant (t Stat parameter is less than 0 (−0.5883), and P(T ≤ t) one-sided (0.2792) is greater than the assumed α = 0.05) (Table 5).

Figure 8 shows a comparison, in the form of a box plot, of the statistical analysis distribution of the patients’ center of mass (measured in centimeters) before therapy (B) and after therapy (A) without the SPIDER system. The range of Box A is within the range of Box B; the ranges overlap, indicating a slight reduction in the tilt of the patients’ center of mass after therapy. A difference of 2% (0.08 cm) between the mean value of the tilt before rehabilitation (Box B: median = 3.55 cm) and after rehabilitation (Box A: median = 3.47 cm) also indicates a slight reduction in the tilt of the body’s center of mass.

The mean difference in tilt measured in centimeters using the TYMO^®^ platform after rehabilitation and measurements before rehabilitation is significantly greater if the SPIDER system was used during rehabilitation (t Stat parameter is less than 0 (−9.8534), and P(T ≤ t) one-sided is less than 0.000001, thus less than the assumed α = 0.05) (Table 6).

The difference in the mean tilt of the patient’s center of mass, measured on the TYMO^®^ platform, before and after rehabilitation, with the use of the SPIDER system, is (−2.15 cm). The difference in the tilt of the patient’s mean center of mass, measured on the TYMO^®^ platform, before rehabilitation and after rehabilitation, using therapy without the SPIDER system, is (−0.08 cm). The larger negative difference indicates that, with the use of the SPIDER system, the recorded tilts decreased to a larger extent during rehabilitation compared to the tilts recorded before rehabilitation, compared to not using the SPIDER system in the training.

The following graphs (Figure 9) illustrate the differences between the tilts of the body’s center of mass before and after therapy: Graph A, with the use of the SPIDER system; and Graph B, without the use of this system. Both graphs represent the entire patient population of the SG and CG groups. Comparing them, we can notice a difference in the quantitative range of the scale used: patients in the SG group achieved results in the range of 0.5–(−6) (cm), whereas patients in the CG group achieved results in the range of 3–(−3) (cm), which confirms the results of the statistical analysis relating to the difference in the mean tilt of the patient’s center of mass (Table 4, Table 5 and Table 6).

## 4. Discussion

Impaired gait ability after stroke significantly limits patients’ ability to participate in many social activities. Gait speed is considered a predictor of disability severity [37,38,39]. Hence, restoration of locomotor ability is considered a major goal of post-stroke physiotherapeutic treatment, while the ability to transfer body weight to the affected side is an indispensable component of successful therapy [40]. In fact, it influences the improvement of balance and the maintenance of gait ergonomics and economy [41,42,43]. In the current study, the effect of therapy using the SPIDER system on the symmetry of body weight transfer, in a standing position, in post-stroke patients in the chronic phase was examined in relation to therapy that did not use this system. The results of the statistical analysis showed a greater reduction in tilt of the body center of mass in the SG group, relative to the CG group. The mean difference in tilts after rehabilitation and before rehabilitation, in which the SPIDER system was used, was (−2.15 cm), while the mean difference in tilts after rehabilitation and before rehabilitation without the SPIDER system, was equal to (−0.08 cm). The larger negative difference indicates that, with the use of the SPIDER system, the recorded tilts decreased to a greater extent during rehabilitation compared to the tilts recorded before rehabilitation than when the SPIDER system was not used in the training. This study used the TYMO^®^ measurement platform, which enabled an objective, direct, quantitative assessment of the patient. Such an assessment can convert into significant physiotherapy efficiencies and consequently an improvement in the patient’s quality of life, as well as contributing to lower rehabilitation costs and a shorter period of abstinence from work. Platforms used to quantitatively measure the body’s center of mass while standing interpret standing stability and postural control strategies [44]. Roerdink et al. found that the range of movement of the body’s center of mass was greater on the non-affected side relative to the affected side during three conditions of still standing: eyes open, eyes open while performing a dual task and eyes closed [45]. Similar findings were presented by Eng and Chu examining stroke patients in the chronic phase. They found that weight-bearing capacity was greater on the unaffected side and that the tilt of the center of mass was smaller on the affected side, especially when shifting weight forward [46]. De Haart et al. studied the restoration of weight-bearing capacity after stroke and found that even people who had suffered a severe stroke were able to improve the speed and accuracy of weight bearing [47]. The results of Nam et al. suggest that forced weight transfer to the affected lower limb may be an effective method to improve gait ability in post-stroke patients [12]. In addition, Park et al. noted that lateral weight transfer to the affected side is a key element in physiotherapeutic treatment after stroke. During gait training, it should be common practice to assist post-stroke patients to laterally shift their body weight to the hemiparesis lower limb, in order to maintain their body weight on that limb and take a step with the unaffected lower limb [40]. It should be noted that data from the literature suggest that post-stroke patients are able to take more than 50% of their body weight on the affected lower limb [37]. Therefore, the difficulty in restoring symmetry of body weight distribution experienced by many post-stroke patients is more likely to be due to a learned reluctance to take weight on the affected lower limb. Therapy using forced weight bearing on the directly affected lower limb reduces the process of learned disuse of the affected limb and maximizes neuroplasticity to restore the ability and normalize weight bearing on that limb [37,48]. Studies have shown that therapy that improves the ability to symmetrically transfer body weight (equal weight distribution on the affected and unaffected lower limb) leads to increased stability during gait and improves sensation in the affected foot [17,49,50]. This is because the repetitive feedback from the loaded receptors during therapy is used as feedback to strengthen the extensors during gait [17,51]. Thus, forced weight shift to the affected lower limb leads to improvements in gait parameters and ground pressure distribution in post-stroke patients [38]. The results of the studies by Aruin et al. also confirmed that intervention using forced weight shift therapy to the affected side can result in long-term improvements in symmetry of weight distribution and gait speed in post-stroke patients in the chronic phase. In their studies, the shift of the center of mass to the affected side increased by 9.7 percent, while gait velocity increased by 10.5 percent [1,37,52], whereas the present study showed a greater reduction in tilt of the body’s center of mass in therapy using the SPIDER system (centralization of the body’s center of mass), compared to therapy in which the system was not used. Relating this to the studies cited above, we can conclude that also the SPIDER system therapy has a high validity in the physiotherapeutic treatment of post-stroke patients.

## 5. Conclusions

The above results indicate the effectiveness of physiotherapy treatment using the SPIDER system, with regard to the symmetry of body weight distribution in post-stroke patients, compared to therapy in which the system was not used. In the study group, there was a marked reduction in the tilt of the body’s center of mass to the unaffected side.

## Figures and Tables

**Figure 1 ijerph-19-16214-f001:**
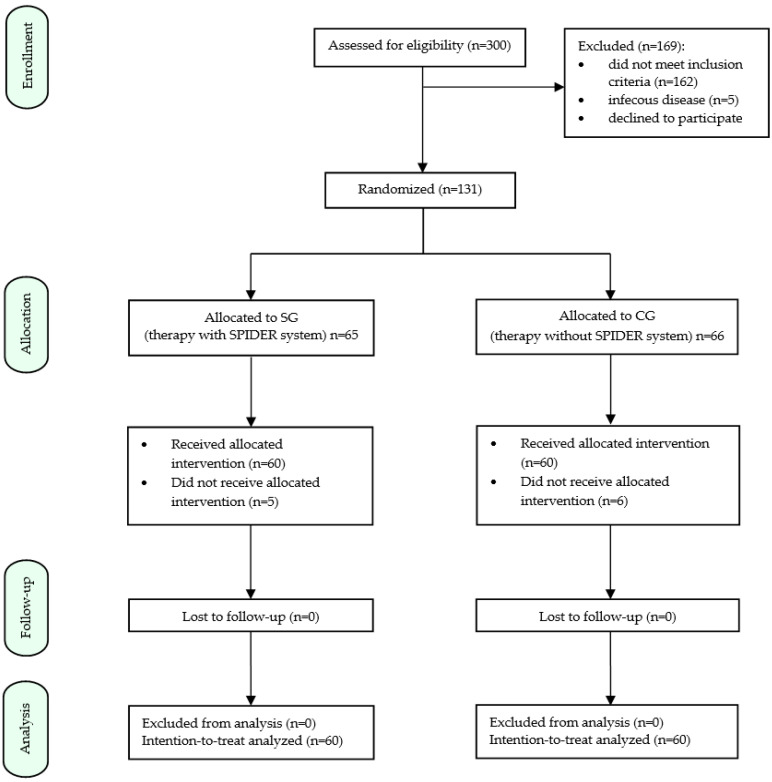
Study design flowchart.

**Figure 2 ijerph-19-16214-f002:**
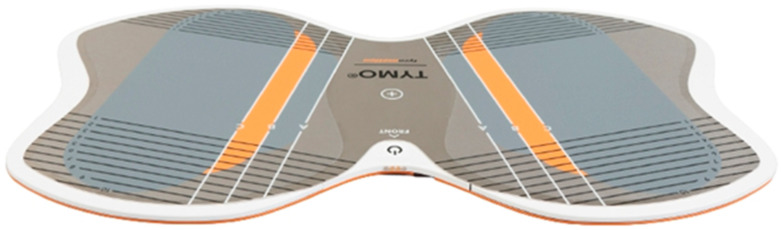
Posturographic TYMO^®^ platform [33].

**Figure 3 ijerph-19-16214-f003:**
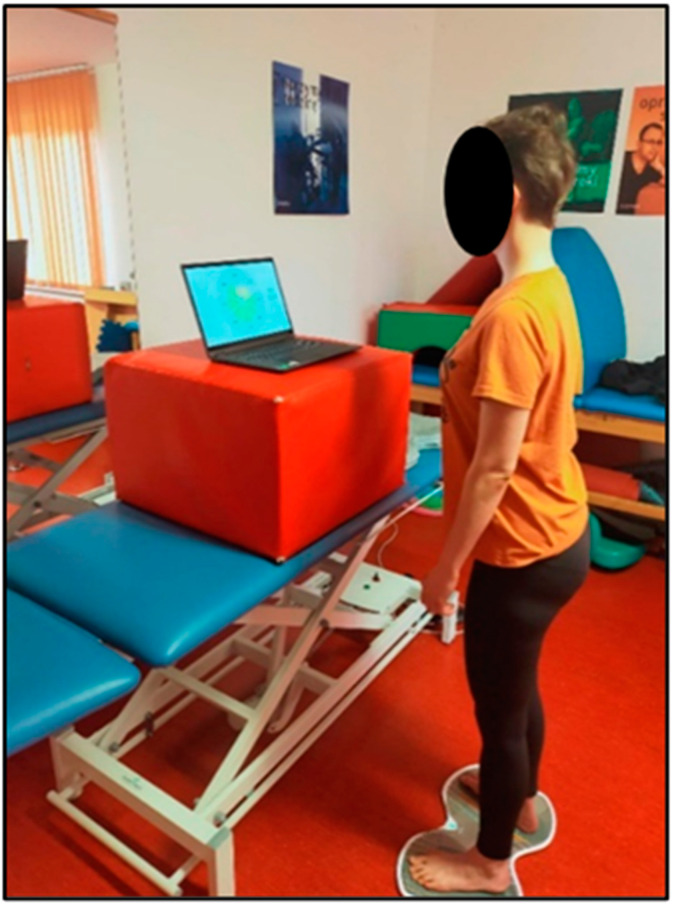
Performing measurements on the TYMO^®^ platform (own source).

**Figure 4 ijerph-19-16214-f004:**
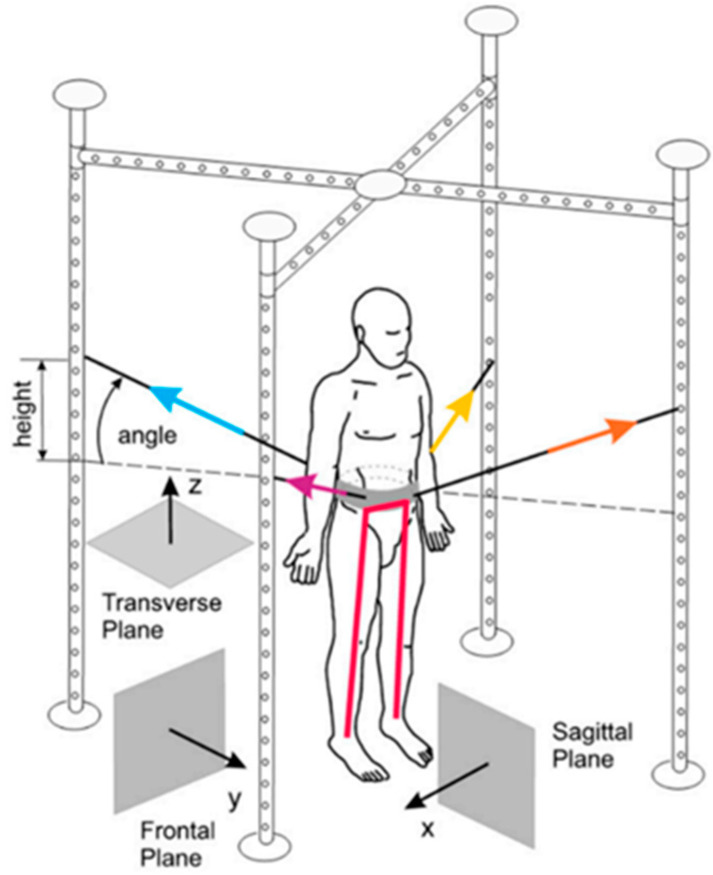
Strengthening Program of Intensive Developmental Exercises and Activities for Reaching Maximal Potential (SPIDER) equipment (SPIDER net) with the reference planes and the expander forces in the standard anatomical position [36].

**Figure 5 ijerph-19-16214-f005:**
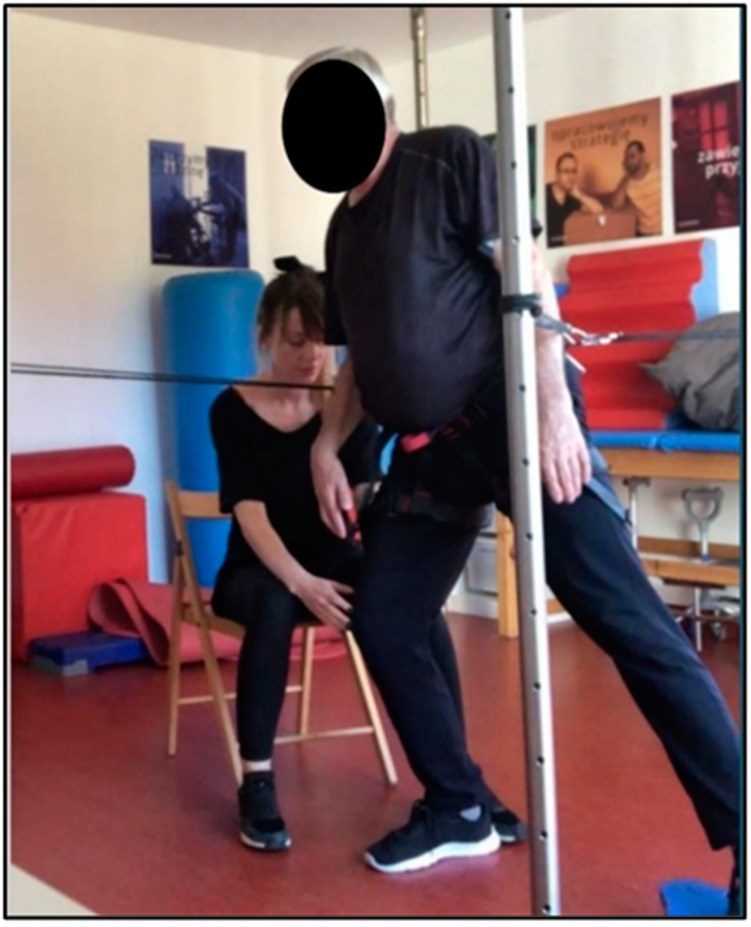
Patient I during therapy with usage of the SPIDER system (own source).

**Figure 6 ijerph-19-16214-f006:**
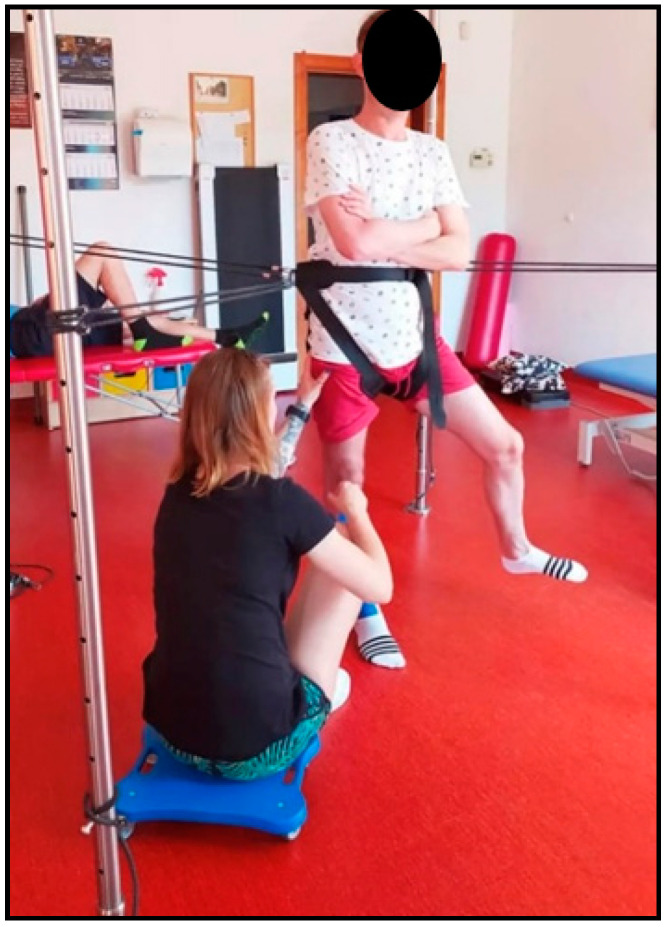
Patient II during therapy with usage of the SPIDER system (own source).

**Figure 7 ijerph-19-16214-f007:**
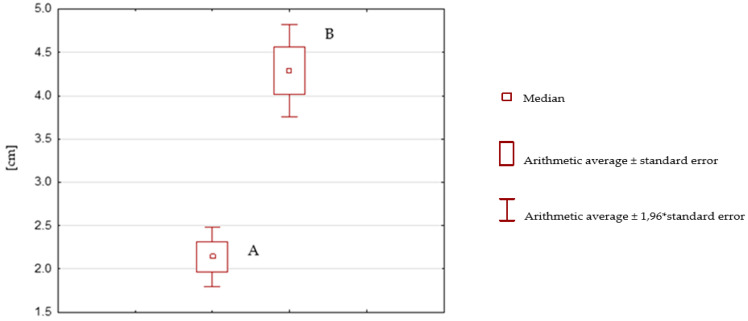
Box plot after rehabilitation with SPIDER system (A) versus before rehabilitation with SPIDER system (B).

**Figure 8 ijerph-19-16214-f008:**
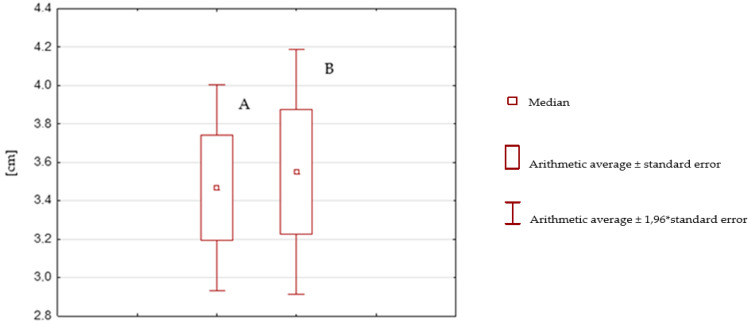
Box plot after rehabilitation without SPIDER system (A) versus before rehabilitation without SPIDER system (B).

**Figure 9 ijerph-19-16214-f009:**
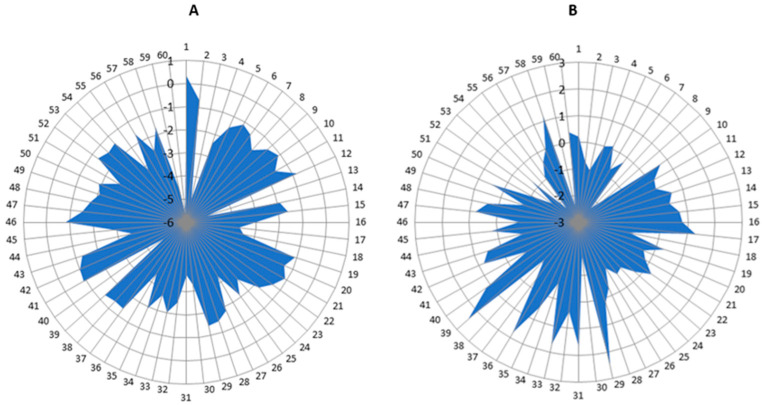
A graph showing the difference post-pre therapy: (**A**) with the SPIDER system; (**B**) without the SPIDER system. 160—number of participants; 2.5–(−6)—amount of deviation.

**Table 1 ijerph-19-16214-t001:** Summary of inclusion and exclusion criteria for the study and control groups.

Inclusion Criteria for the Study and Control Group:
Age 18+. Past ischemic stroke (first episode) as diagnosed by a doctor First 5 years after stroke (chronic phase) Hemiparesis Ability to move from sitting to standing (assessed by functional test) Ability to maintain standing position for the duration of the examination (assessed by functional test) FMA-LE score ≥ 16 points Lower limb spasticity (rectus femoris muscle, ischiofemoral muscle group, gastrocnemius muscle) ≤ 2 points according to the Modified Ashworth Scale NIHSS score ≤ 15 points MMSE score ≥ 24 points BBS ≥ 21 points Modified Rankin Scale ≤ 3 Barthel scale ≥ 55 Informed agreement of the patient to participate in the trial after reading the study document in detail. In case of a patient with aphasia, consent of family or legal guardian.
Exclusion criteria for the study and control group:
Age below 18 years Cerebral hemorrhagic stroke Confirmed second or subsequent stroke Acute phase post-stroke or time since stroke greater than 5 years Lower limb amputation Comorbidities, diagnosed by a doctor, which may have a direct impact on therapeutic procedures, such as apalous state, cancer Diagnosis of TIA in a patient Medical diagnosis assessing the state of depression Medical diagnosis assessing cognitive impairment Inability to stand FMA-LE score < 16 points Lower limb spasticity (straight thigh muscle, ischiofemoral muscle group, gastrocnemius muscle) > 2 points according to the Modified Ashworth Scale NIHSS score > 15 points MMSE < 24 pts. BBS < 21 pts. Modified Rankin Scale > 3. Barthel Scale < 55 Physician-diagnosed global aphasia Use of botulinum toxin within 3 months of the start of the therapeutic program or during the program Taking spasmolytic medication (e.g., Baclofen) during the program Lack of informed consent from the patient for the implementation of therapy procedures.

**Table 2 ijerph-19-16214-t002:** Demographic and functional characteristics of the study participants.

	Average/Value	Standard Deviation	Minimum	Maximum
Gender: F (female)	38 (31.67%)			
M (male)	82 (68.33%)			
Age	56.13	14.4	24	84
Hemiparesis side: R (right)	56 (46.6%)			
L (left)	64 (53.4%)			
Modified Ashworth Scale: 2	112			
1+	8			
Barthel Scale	70	7	55	95
Superficial sensation: N (hyposensitivity)	115			
P (hypersensitivity)	5			
Deep sensitivity: (+/−)	50			
(+)	64			
(−)	6			
BBS	25	2	21	32
NIHSS	7	2	3	11
Rankin Scale	3	0	3	3
FMA-LE	17	1	16	23

**Table 3 ijerph-19-16214-t003:** Demographic and functional characteristics of SG and CG.

	SG				CG			
	Average/Value	Standard Deviation	Minimum	Maximum	Average/Value	Standard Deviation	Minimum	Maximum
Gender: F	18 (30%)				20 (33.33%)			
M	42 (70%)				40 (66.67%)			
Age	59	14.7	24	80	57.5	14.3	25	84
Hemiparesis side: R	29 (48.33%)				29 (48.33%)			
L	31 (51.67%)				31 (51.67%)			
Modified Ashworth Scale: 2	56				56			
1+	4				4			
Barthel Scale	65	6.5	55	85	70	8.1	55	95
Superficial sensation: N	57				58			
P	3				2			
Deep sensitivity: (+/−)	24				26			
(+)	32				32			
(−)	4				2			
BBS	25	1.9	21	30	25	2.2	21	32
NIHSS	7	1.4	4	11	7	2	3	11
Rankin Scale	3	0	3	3	3	0	3	3
FMA-LE	16	0.9	16	20	18	1.5	16	23

**Table 4 ijerph-19-16214-t004:** The results of the statistical analysis of the average tilt before and after rehabilitation with the SPIDER system.

	After Rehabilitation with the SPIDER System	Before Rehabilitation with the SPIDER System
Average	2.14	4.29
Variation	1866169492	4426
Population	60	60
Pearson correlation	0.835873939	
df	59	
t Stat	−13.65422458	
P(T ≤ t) one-sided	3.22093 × 10^−20^	
One-sided *t*-test	1.671093033	
P(T ≤ t) bilateral	6.44186 × 10^−20^	
Two-sided *t*-test	2.000995361	

**Table 5 ijerph-19-16214-t005:** The results of the statistical analysis of the average tilt before and after rehabilitation without the SPIDER system.

	After Rehabilitation without the SPIDER System	Before Rehabilitation without the SPIDER System
Average	3.468333333	3.55
Variation	4.45169209	6.331016949
Population	60	60
Pearson correlation	0.906659242	
df	59	
t Stat	−0.58833267	
P(T ≤ t) one-sided	0.27927728	
One-sided t-test	1.671093033	
P(T ≤ t) bilateral	0.55855456	
Two-sided t-test	2.000995361	

**Table 6 ijerph-19-16214-t006:** The results of the statistical analysis of the average tilt after rehabilitation with and without the SPIDER system.

	After rehabilitation with the SPIDER system	After rehabilitation without the SPIDER system
Average	−2.15	−0.081666667
Variation	1.487627119	1.15609887
Population	60	60
Total variance	1.321862994	
df	118	
t Stat	−9.853441355	
P(T ≤ t) one-sided	2.23299 × 10^−17^	
One-sided *t*-test	1.657869523	
P(T ≤ t) bilateral	4.46597 × 10^−17^	
Two-sided *t*-test	1.980272226	

## Data Availability

Not applicable.

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
