# Peer review of "Evaluation of the Effect of SPIDER System Therapy on Weight Shifting Symmetry in Chronic Stroke Patients—A Randomized Controlled Trial"

_ijerph, 2022, doi:10.3390/ijerph192316214_

Round 1

Reviewer 1 Report

The work is written very correctly, valuable especially in terms of the assessment of new, not very widely known methods of rehabilitation methods. The work is suitable for publication in the journal, but it is suggested to revise it taking into account the following substantive and editorial issues.

1. In the Introduction, enrich the information from the literature and add what methods are used to test: balance and gait regularity after a stroke. For example, next to the cited works 10-12. .

"Gait analysis using a shoe-integrated wireless sensor system."

,,Inertial sensors and wavelets analysis as a tool for pathological gait identification”

2. In Chapter 2 Materials and Method, information (even estimates) should be provided on the population from which the research sample is selected. For example, it should be a random sample for patients from a specific country, region, hospital, etc. At the same time, this should result in the assessment of the sample size n=120 and the statement for which population it is sufficient. In this way, the authors will avoid the accusation that this is a too small sample, so that in Conclusion in the last part they do not have to write a trivial statement: "more research needs to be done on therapy using the SPIDER system, as few papers are available on the subject they would only state for which populations they are relevant.

3. In subchapter 2.4. Statistical analysis authors wrote that: "Differences between pairs of measurements did not show the normality of the distribution". What differences did they consider? They should examine the "normality" of the examined feature of the distribution/population. Or simply knowing what process, phenomenon they are studying, assume that the feature has a normal or close to normal distribution. The sample size does not determine the normality of the distribution from which it was taken.

In terms of editorial:

1. Improve the editorial quality of the tables. Table headers are too large.

2. Pay attention to the correctness of citations. For example, under figure 4.

3. Check the correctness of the cited literature in the bibliography.

4. The block diagram in Figure 1 is built on the principles created by the authors. (inconsistent with the principles of creating flowcharts for algorithms) There should be decision blocks there. At least the Inclusion/Exclusion and Allocation decision block. But to leave this drawing in the form proposed by the authors, the Enrollment block should be attached/linked to it. It is, after all, an element of  The Study Design".

5. Avoid unclear, unexplained statements in the text, such as, for example, "metabolic and biomechanical costs" or "indirectly affected side".

Reviewer 2 Report

Manuscript entitled: "Evaluation of the effect of SPIDER system therapy on weight shifting symmetry in chronic stroke patients – a randomised controlled trial" is an interesting research article. It could be published after revision.

GENERAL CONSIDERATION:

- the acronyms PNF and NDT relatives to the physiotherapy techniques have to be explained both in the abstract and in the main text, because these acronyms aren't cited in the "Abbreviations" form at the end of the manuscript.

- TYMO is a commercial name, so I suggest you to modify in TYMO® in the abstract and in the main text

- I suggest you to reframe "the improvement process" or "the improvement program" (line 26, 141, 143, 167) using a more neutral word as "training" 

INTRODUCTION:

- The muscular changes that occur after a stroke (spasticity in the unaffected side and paresis in the affected side) must be better explained.

MATERIALS AND METHODS

- You have written that the Rankin scale is a five-point scale, explaining the meaning of the scores 0 and 5. In this case, this must be considered as a six-point scale.

- Exclusion criteria must be added: for example, the concomitant use of medications was taken into consideration?

RESULTS:

In the Table 3 there isn't a comparison (expressed with a p value) between the two groups about every demographic/functional characteristic at the baseline. This value must be expressed in the Table, or in alternative, the sentence at the line 239 must be modified as follows "There were no significant differences in demographic and functional values between the groups (for all the variables: p value >0.05)"

Round 2

Reviewer 2 Report

The manuscript is now clearer and more complete, also thanks to the suggestions of the other reviewer.

The Authors have done a great work and the article can be published, hoping that this low-cost and easy to perform therapeutic (SPIDER system) and diagnostic (TYMO® platform) tool will be used in further studies